# MACK: Multimodal Aligned Conceptual Knowledge for Unpaired Image-text Matching

**Yan Huang**      **Yuming Wang**      **Yunan Zeng**      **Liang Wang**
Center for Research on Intelligent Perception and Computing (CRIPAC),
State Key Laboratory of Multimodal Artificial Intelligence Systems,
Institute of Automation, Chinese Academy of Sciences (CASIA)
School of Artificial Intelligence, University of Chinese Academy of Sciences (UCAS)
Chinese Academy of Sciences Artificial Intelligence Research (CAS-AIR)
`{yhuang, wangliang}@nlpr.ia.ac.cn,`
`{yuming.wang, yunan.zeng}@cripac.ia.ac.cn`

## Abstract

Recently, the accuracy of image-text matching has been greatly improved by multimodal pretrained models, all of which are trained on millions or billions of paired images and texts. Different from them, this paper studies a new scenario as unpaired image-text matching, in which paired images and texts are assumed to be unavailable during model training. To deal with this, we propose a simple yet effective method namely Multimodal Aligned Conceptual Knowledge (MACK), which is inspired by the knowledge use in human brain. It can be directly used as general knowledge to correlate images and texts even without model training, or further fine-tuned based on unpaired images and texts to better generalize to certain datasets. In addition, we extend it as a re-ranking method, which can be easily combined with existing image-text matching models to substantially improve their performance.

## 1   Introduction

Image-text matching is one of the most representative techniques in the field of vision and language understanding, which has wide applications such as online shopping, human-robot interaction and autonomous driving. Its major challenge lies in how to accurately measure the cross-modal similarity between images and texts. Recently, by training on very large-scale (millions or billions) paired images and texts, various multimodal pretrained models [23, 20, 7] alleviate the challenge in a supervised learning manner. Although the accuracy of image-text matching is greatly improved, collecting and annotating such large-scale data in real world applications is time-consuming and expensive.

Different from them, this work attempts to study the image-text matching in the context of a new scenario, namely *unpaired image-text matching*, in which paired images and texts are assumed to be unavailable during model training. It is motivated by the fact that the human brain can well correlate arbitrary images with texts while does not need to learn from such large-scale paired images and texts. Instead, it stores semantic knowledge about objects, actions, attributes, *etc.*, which is multimodal aligned and can be used to correlate visual and linguistic information [28, 2]. Inspired by these neuroscience evidences, this work tries to deal with the unpaired image-text matching by modeling human brain-like knowledge.

Unimodal visual or linguistic knowledge has been widely used for vision and language understanding such as image captioning [39], visual reasoning [45] and visual question answering [31]. By directly combining these two types of knowledge together, multimodal knowledge shows more complementary

36th Conference on Neural Information Processing Systems (NeurIPS 2022).

advantages [46, 37]. However, existing knowledge-based methods are always designed to corporate with paired data during model training, which cannot well handle the unpaired image-text matching. Another alternative is multimodal knowledge graphs [47], most of which are extended from existing linguistic knowledge graphs by linking semantically related images to corresponding words. However, the image-word alignment might have the following issues: 1) images usually include redundant or word-unrelated contents, so the images and words carry unequivalent semantic information, and 2) word-related objects (or attributes) usually have diverse appearances in different images, so the alignment is one-to-many rather than one-to-one. In fact, obtaining well-aligned multimodal knowledge is a great challenge.

In this work, we propose a new method namely Multimodal Aligned Conceptual Knowledge (MACK) for unpaired image-text matching. To remove the word-unrelated content in images, it focuses on semantic concepts and collects pairs of words and semantically related image regions from public available datasets. Then it computes prototypical region representations and aligns them to the words, with the goal to alleviate the appearance variant. Based on the aligned conceptual knowledge (*i.e.*, word-prototypical region pairs), the MACK can bridge images and texts in the same feature space to measure their cross-modal similarities. To make the pre-computed general knowledge better suit certain datasets, we can further fine-tune it with the principle of region-level cycle-consistency, which does not need paired images and texts. Since the proposed MACK is simple yet effective, it can well corporate with existing image-text matching models as a re-ranking method to further improve their performance.

Our contributions are summarized as follows. 1) We study a rarely investigated problem as unpaired image-text matching, which is under great demand in practical applications. 2) We accordingly propose a simple yet effective method to model multimodal aligned conceptual knowledge, which can either perform test directly or perform training based on unpaired data. 3) The proposed method can be easily adopted by existing image-text matching models to re-rank their results to obtain better performance.

## 2 Related Work

### 2.1 Image-Text Matching

Image-text matching draws much attention recently, and many effective models trained on paired images and texts have been proposed. Socher *et al.* [32] and Frome *et al.* [6] propose the early framework of Visual-Semantic Embedding (VSE) to correlate images and their labels. Kiros *et al.* [16] extend the VSE from image-label matching to image-text matching, which is later improved from various aspects such as adding intra-modal constraint [36], mining hardest negative sample [5], analyzing canonical correlation [41], matching local instance [11, 13], improving sentence representation [29, 19], and reasoning visual relation [22, 14]. In addition to the VSE, another important framework is Cross-Modal Attention (CMA) proposed by Lee *et al.* [18], which uses pretrained object-level features [1] and then generates cross-modal attended representations for similarity measurement. The CMA is later improved from the directions of cross-modal memory [10, 9], cross-modal message passing [38], hybrid learning [8], context modeling [44], iterative matching [3], and graph structure [25]. Later, many more models [4, 20, 27, 23] based on multimodal versions of Transformer [35] are proposed, which rely on millions or billions of paired images and texts for model training and achieve very good performance. Recently, Huang *et al.* [12] make the early attempt to study how to improve the efficiency of such large image-text matching models, which achieves promising results. In contrast to the existing models, we attempt to study a new scenario of unpaired image-text matching that assumes paired images and texts are unavailable for model training, which is more challenging than conventional image-text matching.

### 2.2 Multimodal Knowledge

There are only a few works that use multimodal knowledge for vision and language understanding tasks. For example, Zhu *et al.* [46] build heterogeneous graphs corresponding to visual, semantic and factual features for visual question answering. Wang *et al.* [37] combine visual and textual knowledge to find discriminative parts of objects and improve the performance of few-shot learning. Zhang *et al.* [43] propose a concept-relation graph and then leverage it for visually grounded concept learning. In addition, different multimodal knowledge graphs [47, 26, 33] are proposed, most of

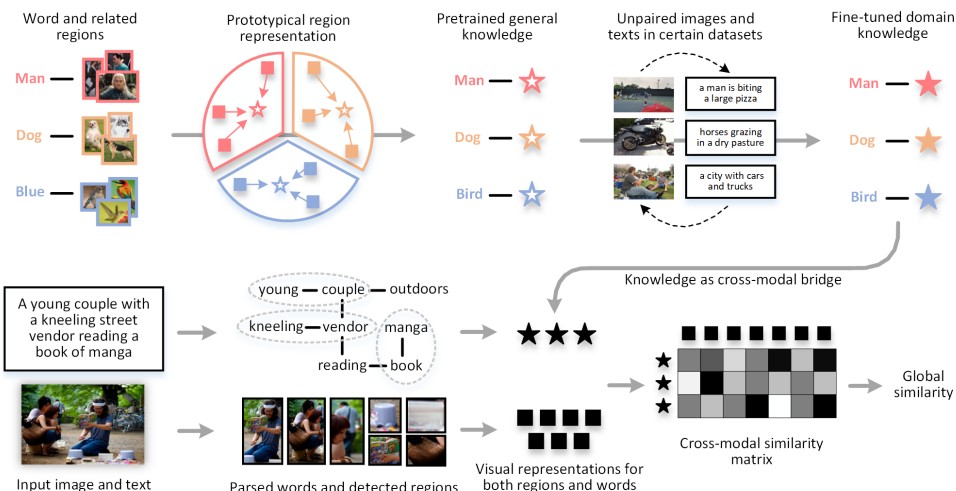

Figure 1: The proposed Multimodal Aligned Conceptual Knowledge (MACK) for unpaired image-text matching. The top figures illustrate how to obtain the knowledge in three steps: 1) collecting a set of conceptual words and their related image regions, 2) computing prototypical region representations to get the pretrained general knowledge, and 3) training on unpaired images and texts to obtain the fine-tuned domain knowledge. The bottom figures illustrate how to use the knowledge for unpaired image-text matching in three steps: 1) obtaining parsed words and detected regions from the input image and text, 2) aggregating related words and representing them by prototypical region representations in the knowledge, and 3) comparing word and region representations in the same feature space and pooling their cross-modal matrix to obtain a global similarity score.

which extend linguistic knowledge graphs by aligning images to words. Such cross-modal alignment could be quite noisy because images usually include redundant or word-unrelated contents. In fact, the multimodal knowledge used in these methods is either cross-modal unaligned or noisy aligned, which usually needs paired images and texts as strong supervisions for model training. Different from them, our multimodal aligned conceptual knowledge establishes more accurate one-to-one alignment in the fine-grained concept-level, which can handle the unpaired image-text matching without using strong supervisions.

## 3   Multimodal Aligned Conceptual Knowledge

As shown in Figure 1, we propose the Multimodal Aligned Conceptual Knowledge (MACK) for unpaired image-text matching. To obtain the knowledge, we first collect a set of conceptual words and their semantically related image regions from public datasets. For each word, we obtain a prototypical region representation by averaging the representations of all related regions. Thus, we obtain the pretrained general knowledge containing pairwise words and prototypical region representations. To make the knowledge better adapt to certain datasets, we then fine-tune the pretrained general knowledge by modeling the concept-level cycle consistency based on unpaired images and texts.

Give an image and a text, we first perform text parsing to obtain a word graph, and use the knowledge as a cross-modal bridge to obtain a set of prototypical region representations for all the words. Then, we perform object detection on the image to obtain another set of region representations. Thus, the representations of words and regions are in the same feature space, in which they can be directly compared to get a cross-modal matrix. After pooling the values in row and column dimensions based on heuristic rules, we can finally obtain the global similarity score between the image and text.

### 3.1   Pretrained General Knowledge

The studied knowledge in this work has two major properties as follows. 1) Concept-level knowledge: the knowledge is built upon two major types of semantic concepts, which correspond to objects and attributes in images, and nouns and adjectives in texts. It is more fine-grained than existing multimodal knowledge graphs that roughly link global images to words. 2) One-to-one alignment:

rather than align a word to multiple possible images in a one-to-many manner, the knowledge focuses on cross-modal one-to-one alignment, with the goal to alleviate the appearance variations of objects and attributes.

We define the knowledge as a set of semantic concepts having paired multimodal representations $\{(w_k, v_k)\}_{k=1,...,K}$, where $w_k \in \mathbb{N}^K$ and $v_k \in \mathbb{R}^F$ are the one-hot word representation and real-valued region representation of $k$-th semantic concept, respectively, and $K$ is the total number of semantic concepts. To maximize the number of concepts, we collect all the words and their related regions in the Visual Genome (VG) dataset [17]. We could alternatively use other datasets, but the VG has more diverse content that is useful for improving the generalization ability of the knowledge. For each word, we can easily obtain its semantically related image regions in the dataset, and different regions usually have diverse visual appearances. To avoid the impact of appearance variations, we compute a prototypical region representation $v_k$ for each word $w_k$ by averaging all its related region representations $\{r_j\}_{j=1,...,J_k}$:

$$v_k = \frac{1}{J_k} \sum\nolimits_{j=1}^{J_k} r_j \tag{1}$$

where each region representation $r_j$ is obtained by feeding a bounding box and an image into the pretrained object detection model Faster-RCNN[1] [1]. Note that the numbers of regions for different words $\{J_k\}_{k=1,...,K}$ are quite imbalanced, which vary from tens to tens of thousands. Intuitively, frequently appeared words such as "man" and "dog" will have more related regions, and their corresponding prototypical region representations might be more robust than those of few-shot words such as "otter" and "nun". However, the few-shot words can still compute initial prototypical region representations with very limited numbers (*e.g.*, $< 20$) of regions and then refine them when more related regions are encountered.

## 3.2 Fine-tuned Domain Knowledge

To better adapt the pretrained general knowledge to certain datasets, we could further fine-tune the prototypical region representations based on the principle of concept-level cycle consistency, if unpaired images and texts in those datasets are available. In particular, given a batch of unpaired images and texts, we first obtain a set of detected regions using the Faster-RCNN above and a set of tokenized words. For the set of words, duplicated ones are removed and the rest words are $\{w_m\}_{m=1,...,M}$. Then all the words can be represented by the corresponding prototypical region representations as $\{v_m\}_{m=1,...,M}$ or $V \in \mathbb{R}^{F \times M}$. For the set of regions, we compute similarity scores between the prototypical region representations (in the pretrained general knowledge) and the set of region representations, and then remove the redundant regions belonging to same concepts in a similar way as Non-Maximum Suppression (NMS). The resulting regions are denoted as $\{r_n\}_{n=1,...,N}$ or $R \in \mathbb{R}^{F \times N}$.

To fine-tune the prototypical region representations to better suit for certain datasets, we design a cycle consistent loss to learn a parametric transformation matrix $W \in \mathbb{R}^{F \times F}$. The main idea is to first measure similarities between each word and all regions, and use the similarities as weights to combine all regions to obtain a reconstructed word[2]:

$$\hat{V} = WR\sigma(S)^T, \; S = (WV)^T WR \tag{2}$$

where $S \in \mathbb{R}^{M \times N}$ is the similarity matrix between transformed word and region representations, $\sigma(\cdot)$ is the softmax operation along the column dimension, and $\hat{V} \in \mathbb{R}^{F \times M}$ contains the reconstructed word representations. Then, each original word and its reconstructed one are compared to generate a label indicating whether they are the same or not. By minimizing the cross-entropy loss $L$ based on the predicted label and groundtruth one, we can optimize the $W$:

$$L = -\sum\nolimits_{m=1}^{M} y_m^T \log(\hat{y}_m), \; \hat{Y} = \sigma(\hat{V}^T(WV))^T \tag{3}$$

where $\hat{Y} \in \mathbb{R}^{M \times M}$ includes the predicted labels, in which each column is denoted as $\hat{y}_m$, and $y_m$ is the groundtruth label in which the $m$-th value is one and the rest values are zeros. After the training, we can use the $W$ to transform all prototypical region representations to obtain the fine-tuned domain knowledge, denoted as $\{(w_k, u_k)\}_{k=1,...,K}$, where $u_k = Wv_k \in \mathbb{R}^F$.

---

[1]We could alternatively use other pretrained models to detect regions. We select this model because it has been widely used and demonstrated to be effective for image-text matching.

[2]In fact, we similarly combine words to obtain reconstructed regions in the reverse direction.

### 3.3 Knowledge-based Unpaired Matching

To decide whether an image and a text is matched or not, we use the knowledge[3] to measure the cross-modal similarity. For the image, we use the Faster-RCNN to obtain a set of detected region representations $\{r_i\}_{i=1,...,I}$ or $R \in \mathbb{R}^{F \times I}$. For the text, we use the knowledge as a cross-modal bridge to represent all the words as a set of prototypical region representations $\{u_j\}_{j=1,...,J}$. Even though these two set of representations are in the same feature space, they have different amount of semantic information, directly comparing their similarity might be sub-optimal. In fact, each $u_j$ is concept-level corresponding a word, while each $r_i$ usually describes not only an object but also its attribute. To make a better comparison, we infer the relation among words and aggregate related word representations before the similarity measurement.

In particular, for the words in the text, we use the Natural Language Toolkit (NLTK)[4] to predict their types and parse their dependencies. Then, we take each noun as an anchor and combine its representation with dependent adjective representation in an averaged manner. Thus, the set of word representations $\{u_j\}_{j=1,...,J}$ are re-computed as $\{u_l\}_{l=1,...,L}$ or $U \in \mathbb{R}^{F \times L}$. Then we can obtain the desired similarity score $s$ for the image and text as:

$$s = \rho(U^T R) \tag{4}$$

where $\rho(\cdot)$ is the max-mean pooling that first performs max pooling on the column dimension and then mean pooling on the row dimension on the input matrix. Given multiple images and texts, we can similarly obtain their similarity scores for unpaired image-text matching.

### 3.4 Re-ranking Existing Models

Note that the knowledge in this work is inspired by human-like knowledge, which is quite different from the pretrained model knowledge learnt from paired data. So the proposed MACK tends to have complementary properties when combining with existing pretrained models. To demonstrate this idea, we try to use the knowledge to re-rank the results by existing models to improve their performance.

Taking the text-driven image retrieval as an example, given a text as query and a set of $G$ images as gallery, an existing model can produce a similarity vector $\mathbf{s} \in \mathbb{R}^{G \times 1}$. By sorting all the scores of $\mathbf{s}$ in the descending order, the model is usually able to rank the matched image in top-$k$. Then we can tune the top-$k$ scores by adding the corresponding scores predicted by the knowledge with a balancing factor $\lambda$, and then re-rank the top-$k$ images to improve the rank of matched image.

### 3.5 Discussion

Note that the proposed MACK defines a general framework for adaptive knowledge representation, organization, and reasoning. For the knowledge representation, it uses the prototype learning to obtain pairwise prototypical region representations and semantic words. Then, it potentially organizes the knowledge in a graph-like manner, in which the dependency relation among different knowledge pairs are explored. At last, it hierarchically reasons the knowledge in three levels, including domain-level by fine-tuning on certain datasets, sample-level by using the pretrained models, and instance-level by performing the max-mean pooling.

## 4 Experimental Results

To demonstrate the effectiveness of the proposed method, we perform extensive experiments of image-text matching on two publicly available datasets.

### 4.1 Datasets and Protocols

The details of datasets and protocols are as follows. 1) Flickr30k [42] consists of 31783 images collected from the Flickr website. Each image has 5 human annotated texts. We use the public training, validation and testing splits, which contain 29000, 1000 and 1000 images, respectively. 2)

---

[3]The knowledge here could be the pretrained general knowledge or fine-tuned domain knowledge.
[4]https://www.nltk.org/.

Table 1: Unpaired image-text matching by ablation models of MACK on the Flickr30k dataset. Details about these ablation models are explained in Section 4.3.

| Method | Flickr30k dataset | | | | | | |
| | Image Annotation | | | Image Retrieval | | | Rs |
| | R@1 | R@5 | R@10 | R@1 | R@5 | R@10 | |
|---|---|---|---|---|---|---|---|
| Pretrained general knowledge | 10.8 | 26.6 | 35.8 | 4.1 | 10.8 | 18.0 | 106.2 |
| - w/o region prototypes | 1.5 | 6.1 | 9.3 | 1.1 | 4.5 | 9.0 | 31.7 |
| - w/o adjectives | 9.8 | 25.0 | 34.8 | 4.2 | 10.9 | 16.8 | 101.5 |
| - w/o max-mean pooling | 4.2 | 13.4 | 20.4 | 2.9 | 8.4 | 12.8 | 62.2 |
| Fine-tuned domain knowledge | 12.7 | 30.4 | 40.8 | 10.3 | 25.1 | 34.0 | 153.5 |
| - w/o implicit prior | 11.5 | 30.1 | 40.6 | 10.5 | 24.5 | 33.9 | 151.1 |
| - w/o knowledge | 0.1 | 0.5 | 0.9 | 0.0 | 0.6 | 1.1 | 3.2 |

MSCOCO [24] consists of 123287 images, each of which is associated with 5 texts. We use the public training, validation and testing splits, with 113287, 1000 and 5000 images, respectively.

The image-text matching usually includes two sub-tasks in terms of: 1) image annotation: retrieving related texts given images, and 2) image retrieval: retrieving related images given texts. The commonly used evaluation criterions are "R@1", "R@5" and "R@10", *i.e.*, recall rates at the top-1, 5 and 10 results. Following other methods, we also use an additional criterion of "Rs" by summing all the recall rates to evaluate the overall performance.

## 4.2  Implementation Details

In the pretrained general knowledge, we collect all the words in the Visual Genome dataset, so the total number of semantic concepts is $K = 27801$. For each image, we use the pretrained Faster-RCNN to extract region representations, so the number of detected regions is $I = 36$ and the dimension of (prototypical) region representation is $F = 2048$. In the fine-tuned domain knowledge, we perform the model training in a minibatch manner. Each minibatch includes unpaired 128 images and 128 texts that are randomly sampled from the whole training set. We use the Adam algorithm [15] to optimize the only parameter matrix $W$ with a learning rate of $2e^{-4}$ for 30 epochs. When re-ranking existing models, we empirically set $k = 15$ and $\lambda = 0.1$. The knowledge-based unpaired image-text matching is very efficient, which does not need GPUs for acceleration.

## 4.3  Unpaired Image-text Matching

Since existing models are all proposed for paired image-text matching, we compare several ablation models of the proposed MACK to verify their effectiveness for unpaired image-text matching. The first ablation model is pretrained general knowledge, which does not need model training. Then we design its three model variants including: 1) replacing the prototypical region representation with a random region representation, similar to existing multimodal knowledge graphs, 2) removing the adjectives when re-computing the word representations, and 3) replacing the max-mean pooling with mean-mean pooling. We also present another ablation model as fine-tuned domain knowledge, which additionally performs a training stage on unpaired images and texts using the concept-level cycle consistent loss. We also design its two model variants including: 1) using Flickr30k images and MSCOCO texts for the fine-tuning rather than Flickr30k images and texts, and 2) directly optimizing the cycle-consistency loss without using the knowledge.

We compare the performance of unpaired image-text matching by all these models on the Flickr30k dataset in Table 1. It is reasonable that the pretrained general knowledge achieves much worse performance than existing image-text matching models trained on paired images and texts. The unpaired image-text matching still needs more research and there is still a lot of room for performance improvement. By removing the region prototypes, adjectives, and max-mean pooling[5], the performance all becomes worse. And we find that the region prototypical representation makes the largest contribution to the performance of pretrained general knowledge. By training on the unpaired

---

[5]In addition to the max-mean pooling, we also performed experiments of mean-max pooling, max-max pooling, and mean-mean pooling, but found they achieve much worse performance.

Table 2: Image-text matching by re-ranking two state-of-the-art models on the Flickr30k and MSCOCO datasets. "+" denotes the re-ranking.

| Method | Flickr30k dataset | | | | | | | MSCOCO dataset | | | | | | |
|---|---|---|---|---|---|---|---|---|---|---|---|---|---|---|
| | Image Annotation | | | Image Retrieval | | | Rs | Image Annotation | | | Image Retrieval | | | Rs |
| | R@1 | R@5 | R@10 | R@1 | R@5 | R@10 | | R@1 | R@5 | R@10 | R@1 | R@5 | R@10 | |
| CLIP [30] | 65.4 | 87.2 | 91.7 | 85.4 | 97.1 | 98.7 | 525.6 | 35.3 | 60.0 | 70.1 | 55.2 | 78.7 | 86.7 | 386.1 |
| + MACK | 66.8 | 88.2 | 92.6 | 86.2 | 97.2 | 98.9 | 530.0 | 36.9 | 61.6 | 71.7 | 55.7 | 79.6 | 87.1 | 392.8 |
| ALBEF [21] | 59.9 | 84.8 | 90.5 | 78.2 | 95.5 | 97.9 | 506.9 | 40.2 | 68.4 | 78.9 | 62.4 | 85.9 | 92.1 | 428.3 |
| + MACK | 61.8 | 85.8 | 91.4 | 80.1 | 96.4 | 97.7 | 513.3 | 41.0 | 69.0 | 79.4 | 62.4 | 86.1 | 92.7 | 430.9 |

| Query text | Returned top-3 images by CLIP | Query text | Returned top-3 images by CLIP |
|---|---|---|---|
| woman wearing a yellow hat pink shirt and red apron is holding food in a kitchen | | a group of people are sitting on the grass outside of a rustic building | |
| man and woman wearing sunglasses sit halfway in the water | | two people in blue shirts are outside with a bullhorn | |

Figure 2: Examples of retrieved top-3 images based on text queries by the CLIP. Groundtruth matched images are marked by red bounding boxes, which can be re-ranked higher by the MACK.

images and texts, the fine-tuned domain knowledge shows significant performance improvements, especially on the sub-task of image retrieval. By additionally removing the implicit prior that there exist one-to-one correspondences between images and texts, the results become slightly worse but are still much better than pretrained knowledge. The results are very bad if we do not use the knowledge as the cross-modal bridge to correlate images with texts. In the following experiments, we use the pretrained general knowledge as the default MACK due to its simplicity.

## 4.4 Re-ranking State-of-the-art Models

The recent state-of-the-art models for image-text matching are variants of multimodal Transformers, all of which are pretrained on very large-scale datasets containing millions or billions of paired images and texts. To demonstrate that our knowledge-driven MACK is complementary with existing data-driven models, we use the MACK to re-rank the retrieval results of two state-of-the-art models including CLIP [30] and ALBEF [21][6]. The original and re-ranked performance of these two models on the Flickr30k and MSCOCO datasets are shown in Table 2. It should be noted that, we use the zero-shot versions of these two models with the goal to make fair comparisons with the proposed MACK. The zero-shot means that they are pretrained on large-scale external datasets while directly perform test on the Flickr30k and MSCOCO datasets. From the table, we can see that although their original accuracies are very high, re-ranking with our proposed MACK can further improve them. For the CLIP, the re-ranking can lead to 1.4% and 1.6% improvements in R@1 of image annotation on the two datasets, respectively. For the ALBEF, the re-ranking can lead to overall 6.4% improvement in Rs on the Flickr30k dataset.

These evidences demonstrate that the MACK does show complementary properties when combining with existing models. To further explain this, we show some examples of retrieved images based on text queries by the CLIP in Figure 2. These examples are selected based on the following rules: 1) ranks of groundtruth matched images (marked by red bounding boxes in the figure) by the CLIP are not top-1, and 2) our proposed MACK can re-rank these matched images higher. In the first example, it seems that the CLIP cannot well understand the "yellow hat", "red apron" or "pink shirt" in the query text, since the first retrieved image does not contain either of these semantic concepts but ranks

---

[6]Note that the original ALBEF is a hybrid model that includes both dual-stream architecture and single-stream architecture. Here we only use the dual-stream one with the goal to make a fair comparison with the two-stream CLIP.

Table 3: Cross-dataset image-text matching by re-ranking existing models on the Flickr30k and MSCOCO datasets. "+" denotes the re-ranking. MSCOCO -> Flickr30k: training existing models on the MSCOCO dataset and testing them on the Flickr30k dataset. Flickr30k -> MSCOCO: training existing models on the Flickr30k dataset and testing them on the MSCOCO dataset.

| Method | MSCOCO -> Flickr30k | | | | | | | Flickr30k -> MSCOCO | | | | | | |
| | Image Annotation | | | Image Retrieval | | | Rs | Image Annotation | | | Image Retrieval | | | Rs |
| | R@1 | R@5 | R@10 | R@1 | R@5 | R@10 | | R@1 | R@5 | R@10 | R@1 | R@5 | R@10 | |
|---|---|---|---|---|---|---|---|---|---|---|---|---|---|---|
| VSRN [30] | 42.3 | 69.3 | 78.1 | 53.1 | 79.5 | 87.5 | 409.9 | 14.0 | 31.7 | 42.2 | 20.4 | 40.0 | 50.0 | 198.4 |
| + MACK | 42.6 | 69.8 | 78.5 | 53.3 | 79.7 | 87.7 | 411.7 | 14.4 | 32.6 | 43.1 | 20.5 | 40.5 | 50.2 | 201.4 |
| SAEM [40] | 41.4 | 70.2 | 80.0 | 53.4 | 80.9 | 89.6 | 415.5 | 14.8 | 34.0 | 45.0 | 23.2 | 45.4 | 57.4 | 219.8 |
| + MACK | 41.8 | 70.7 | 80.0 | 54.2 | 81.2 | 89.9 | 417.9 | 15.4 | 34.9 | 45.9 | 23.6 | 46.0 | 57.7 | 223.4 |

higher than the groundtruth one. While our MACK especially focuses on these semantic concepts and can improve the corresponding similarity between the matched image and query text.

### 4.5 Cross-dataset Image-text Matching

We also try to re-rank existing image-text matching models when applying them to the scenario of cross-dataset image-text matching. The experimental settings are explained as follows: 1) selecting two representative image-text matching models (VSRN [30] and SAEM [40]) and training them on a source dataset (*e.g.*, Flickr30k or MSCOCO), 2) performing model test on a different target dataset (*e.g.*, MSCOCO or Flickr30k), and 3) using the proposed MACK to re-rank their results on the target dataset. Note that either our MACK or the two models do not use the target dataset for model training.

In Table 3, the results of two directions of cross-dataset generalization, *i.e.*, MSCOCO -> Flickr30k and Flickr30k -> MSCOCO, are both presented. We can find that using the MACK to re-rank the VSRN and SAEM can improve their generalization abilities in the cross-dataset scenario. Although the relative improvements might be a little bit small, they are substantial in all 7 evaluation criterions. Comparing Table 3 with Table 2, it seems that the performance improvements of re-ranking partially depend on the performance of base models. When the base models are more accurate, the performance improvements of re-ranking are larger.

### 4.6 Prototypical Representation Analysis

In the proposed MACK, one of the most important part is the prototypical region representation. Even the same object could have completely different appearances in images, so using how many regions to compute each region prototype is crucial. We count the number of regions for each word in the Visual Genome dataset and show the statistical result in Figure 3, in which the horizontal axis is word index and vertical axis is logarithmic transformed number of regions. From the figure, we can see that about 80% words have very small numbers of regions, *i.e.*, less than 20 (denoted by the dashed line). Therefore, for most semantic concepts in the knowledge, computing their prototypical region representations is data-efficient.

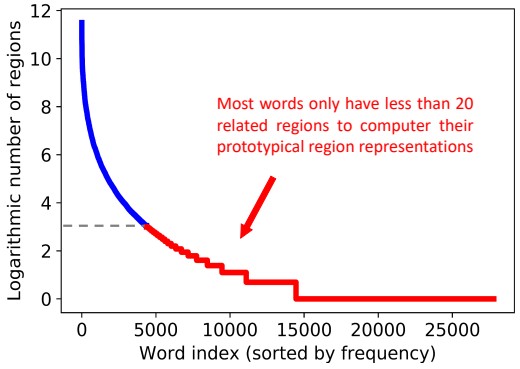

Figure 3: Logarithmic number of regions *v.s.* word indexes. (best viewed in colors).

To verify whether using small numbers of regions has large impact on the performance, we define a threshold $f$ that controls the maximum number of regions. We use the ALBEF + MACK as the base model and select $f$ from {20, 50, 100, $\infty$ }, where $\infty$ is the default setting of ALBEF + MACK. The experimental results on the Flickr30k and MSCOCO datasets are shown in Table 4, in which $p$ is the percentage of how many word-region annotations of Visual Genome dataset are used. We can see that as the $f$ becomes larger

Table 4: Image-text matching by ALBEF + MACK on the Flickr30k and MSCOCO datasets. $f$ is the maximum number of used regions when computing a prototypical region presentation. $p$ is the percentage of how many word-region annotations of Visual Genome dataset are used.

| $f$ | $p$ | Flickr30k dataset | | | | | | | MSCOCO dataset | | | | | | |
|---|---|---|---|---|---|---|---|---|---|---|---|---|---|---|---|
| | | Image Annotation | | | Image Retrieval | | | Rs | Image Annotation | | | Image Retrieval | | | Rs |
| | | R@1 | R@5 | R@10 | R@1 | R@5 | R@10 | | R@1 | R@5 | R@10 | R@1 | R@5 | R@10 | |
| 20 | 15% | 60.0 | 85.1 | 91.2 | 79.7 | 96.0 | 97.7 | 509.8 | 39.7 | 68.0 | 79.1 | 62.2 | 85.6 | 92.7 | 427.5 |
| 50 | 37% | 60.2 | 85.3 | 91.1 | 79.9 | 96.0 | 97.7 | 510.3 | 39.7 | 68.1 | 79.1 | 62.2 | 85.7 | 92.7 | 427.8 |
| 100 | 73% | 60.7 | 85.3 | 91.2 | 80.1 | 95.9 | 97.7 | 510.9 | 39.9 | 68.3 | 79.2 | 62.5 | 86.0 | 92.7 | 428.8 |
| $\infty$ | 100% | 61.8 | 85.8 | 91.4 | 80.1 | 96.4 | 97.7 | 513.3 | 41.0 | 69.0 | 79.4 | 62.4 | 86.1 | 92.7 | 430.9 |

Table 5: Image-text matching by re-ranking CLIP using different $k$ and $\lambda$ on the Flickr30k dataset. $k$ is the length of re-ranked images/texts. $\lambda$ is the balancing factor between knowledge and CLIP.

| $k$ | Image Annotation | | | Image Retrieval | | | Rs | $\lambda$ | Image Annotation | | | Image Retrieval | | | Rs |
|---|---|---|---|---|---|---|---|---|---|---|---|---|---|---|---|
| | R@1 | R@5 | R@10 | R@1 | R@5 | R@10 | | | R@1 | R@5 | R@10 | R@1 | R@5 | R@10 | |
| 10 | 66.8 | 88.2 | 91.7 | 86.2 | 97.2 | 98.7 | 528.8 | 0.1 | 66.8 | 88.3 | 92.7 | 86.2 | 97.2 | 98.9 | 530.1 |
| 15 | 66.8 | 88.3 | 92.7 | 86.2 | 97.2 | 98.9 | 530.1 | 0.2 | 66.6 | 88.6 | 92.9 | 86.4 | 97.1 | 98.8 | 530.5 |
| 20 | 66.8 | 88.3 | 92.7 | 86.2 | 97.2 | 98.9 | 530.1 | 0.3 | 66.6 | 88.4 | 93.0 | 83.5 | 97.0 | 98.7 | 527.1 |

the performance becomes slightly better. It seems that different numbers of regions do not have a significant impact on the performance. It is reasonable since computing the mean might not need too many samples. This is also a good property for the knowledge, since if we want to add a new semantic concept, we only have to annotate a limited number of regions.

## 4.7 Hyperparameter Analysis

In Table 5, we perform the experiment of image-text matching by re-ranking CLIP using different $k$ and $\lambda$ on the Flickr30k dataset, to study the impact of these two hyperparameters on the final performance. In the table, $k$ is the length of re-ranked images/texts, and $\lambda$ is the balancing factor between knowledge and CLIP. In the table, we vary $k \in \{10, 15, 20\}$ and $\lambda \in \{0.1, 0.2, 0.3\}$, and find that the final performance is not very sensitive to the hyperparameters.

## 4.8 Knowledge Visualization

At last, we visualize the pretrained general knowledge and fine-tuned domain knowledge (trained on the Flickr30k dataset) in Figure 4, to illustrate major changes after the training. In the figure, there are two low-dimensional word distributions (surrounded by solid boxes) corresponding to the pretrained general knowledge and fine-tuned domain knowledge, respectively. The two word distributions are obtained by using the t-SNE method [34] to embed two set of prototypical region presentations (i.e., $\{v_k\}$ or $\{u_k\}$, $k = 1,...,$K). For clear illustration, we set $K = 100$ and only select the most frequently used 100 semantic concepts in the Flickr30k dataset. For each semantic concept, we show the corresponding word in the two-dimensional coordinate. We also group semantically related words and show them with different colors.

In the left word distribution, we can see that semantically related words stay nearby. For example, the words in the three groups (marked by dash lines) are about animals, indoor objects and transports, respectively. In the center area, there are many words (marked by the green color) about colors. However, there are still some related words that have remote distances. For example, in the bottom there are also several words about animals such as bird, dog, bear and cat, which are far from the words in the first group. After fine-tuning the knowledge with unpaired data, we find the word distribution is semantically more compact. We can see that all the words about animals are very close staying together in the first group. In the second group, more words about transports such as plane and boat are added. These evidences indicate that by training with the concept-level cycle consistency, related semantic concepts are able to leverage their potential relations to make the prototypical region representations more discriminative.

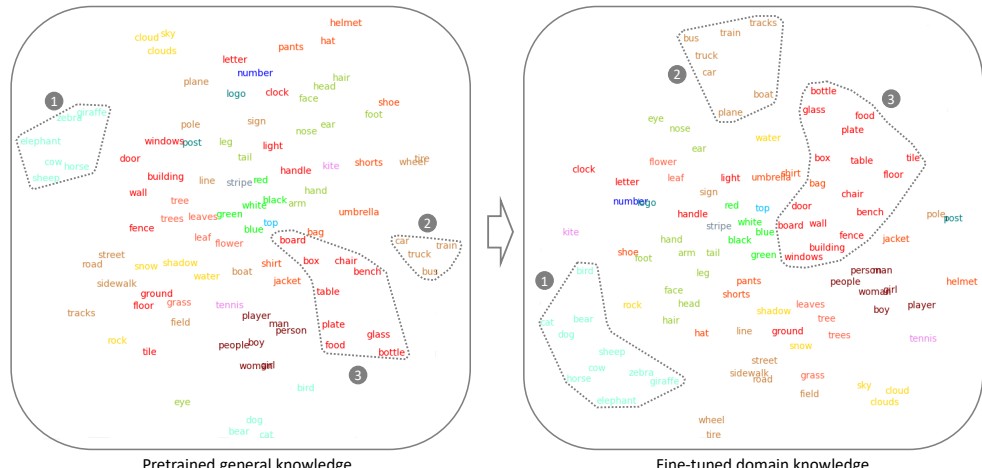

Figure 4: Visualization of pretrained general knowledge and fine-tuned domain knowledge. Each word indicates its corresponding prototypical region representation embedded by the t-SNE method. In the two word distributions, we divide words into semantically related groups and represent them using different colors. We also mark three representative groups by dashed lines, which contain words about animals, indoor objects and transports, respectively.

## 5 Conclusion and Future Work

This work has studied a practically important but seldom investigated problem as unpaired image-text matching, in which paired images and texts are not available during model training. To deal with this problem, inspired by the knowledge used in human brain, we have proposed a simple yet effective method namely Multimodal Aligned Conceptual Knowledge (MACK). Compared with existing multimodal knowledge methods, the MACK has two major properties in terms of conceptual level and one-to-one alignment. It can either perform test directly or perform training based on unpaired data, as well as re-rank existing image-text matching models to improve their performance. We have demonstrated its effectiveness with extensive experiments.

To the best of our knowledge, this might be the first work to study the unpaired image-text matching. So the goal of this initial work is not pursuing high performance but to explore the modeling of knowledge. In the future, we will study more effective strategies to improve the relatively low accuracy. In addition, we will also consider to mine more useful information from the knowledge from the aspects of dynamic knowledge updating and advanced knowledge reasoning.

## Acknowledgements

This work was jointly supported by National Key Research and Development Program of China Grant No. 2018AAA0100400, National Natural Science Foundation of China (62236010, 62276261, 61721004, and U1803261), Key Research Program of Frontier Sciences CAS Grant No. ZDBS-LY-JSC032, Beijing Nova Program (Z201100006820079), and CAS-AIR.

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
