# OpenReview forum: "MACK: Multimodal Aligned Conceptual Knowledge for Unpaired Image-text Matching"
_NeurIPS.cc/2022/Conference — NeurIPS 2022 Accept_

### Official Review · Reviewer_NfVu · 2022-07-07

**Rating:** 6
**Confidence:** 3
**Soundness:** 3 good
**Presentation:** 3 good
**Contribution:** 2 fair

**Summary:**

This paper presents a framework for performing unpaired image-text matching. It leverages the Visual Genome dataset and Faster-RCNN to form a region-based method that matches words and regions of images and promises to yield robust image-sentence-level descriptors for pairing. A cycle consistency loss is proposed to improve performance on specific datasets. Since the proposed method is expected to handle region-word correspondences that current image-text matching models such as CLIP cannot solve well, it can be integrated into current systems and improve their matching performance. The authors tested the proposed method on two common benchmarks, Flickr30k and MSCOCO, and conducted image-text pairing, re-ranking matching results, and cross-dataset matching experiments. Results shown the strength of the proposed method.

**Questions:**

The current hyperparameter used in the re-ranking experiments, $k=15$ and $\lambda =0.1$, seems are set very specific. The reviewer is interesting if the author can do ablations on $k$.

What is the computation cost of the fine-tuned domain knowledge stage? $F$ and $M$ seem too large to make this step practical. If so, the proposed way which involves a transformation matrix is impractical. Does it possible to introduce a NN approximates $W$ but much more efficiently?

Considering the performance of the "pretrained general knowledge", can the author provide the results which only use half the data of Visual Genome? Since annotating Visual Genome is pretty expensive, it is important to see if the proposed approach is sensitive to the sample numbers of the used extra dataset.

**Limitations:**

Clearly, the current method is limited to the visual properties annotated in Visual Genome and the ability of the pre-trained Faster-RCNN. If we want to study fine-grained structures within objects, the proposed method will fail. Also, the method will fail if no object associated with the word is detected.

Considering the proposed framework: (1) mainly relies on the extra fine-grained human-annotated dataset and pre-trained detectors; (2) has very poor results itself; (3) has minor improvements when integrating with the current SOTA models via carefully selected hyper-parameters; (4) the proposed fine-tuned stage is probably impractical, it is more like a proof-of-concept paper. The following researchers are hard to scale up the used fine-grained dataset and find stronger pre-trained detectors, and also, there are web-scale paired image-text datasets. The reviewer is not sure if this paper meets the bar of NeurIPS.

If the mentioned points got addressed to some levels, the reviewer is willing to adjust the scores accordingly.

----------------------------------
After rebuttal: the author has well addressed my concerns and questions. The paper now has ablation studies around the hyperparameters and related discussions about the computation resource and the dataset scale. I raised my scores from 4 to 6.

**Strengths And Weaknesses:**

The studied direction is very important and novel that trying to pair the image and text data according to region-based information. As the author mentioned in the introduction, images usually contain information beyond the info delivered by words. The proposed region-based method hopefully shows more robust performance in conducting image text alignment. All the used components are technically sound, including using Visual Genome, the pre-trained Faster RCNN, and the cycle-consistency loss. Overall the paper is easy to follow.

The author conducted experiments on two common datasets, Flickr30k and MSCOCO, and adopted various baselines, including CLIP, ALBEF, VSRN, and SAEM.  Although the proposed method itself performance is much worse than the models trained on pairs of image-text datasets, it can further improve the performance of those models as shown in Table 2. Also, the reviewer pretty much likes the Cross-dataset experiments, where both the used baseline + MACK did not see the target dataset during training.

According to the gap between Table 1 and Table 2, it is clear that the current method which leverages VG dataset and pre-trained Faster R-CNN is not powerful enough to conduct image-text alignment. The author did not ablate the used hyper-parameters in re-ranking experiments. The current minor improvements shown in Table 2 might come from carefully hyper-parameter searching. So does Figure 2.

---

> ### Author Response · Authors · 2022-08-02
> **Response to Reviewer NfVu**
>
> Thank you for spending so much time and effort on reviewing our paper, we will follow your valuable comments to improve our paper in the revision. Our responses to your concerns are as follows.
>
> 1. The current hyperparameter seems are set very specific. The reviewer is interesting if the author can do ablations on k.
> > Thanks for the reminder. The ablation study of two hyperparameters using the pretrained general knowledge are shown in the following two tables. We will add these experiments and analysis in the revision.
> >>
> >Table: Image-text matching by re-ranking CLIP using different $k$s on the Flickr30k dataset.
> | $k$ | R@1(i2t) | R@5(i2t) | R@10(i2t) | R@1(t2i) | R@5(t2i) | R@10(t2i) | Total |
> | -- | :---: | :---: | :----: | :---: | :---: | :--: | :---: |
> | 10 | 66.8 | 88.2 | 91.7 | 86.2 | 97.2 | 98.7 | 528.8 |
> | 15 | 66.8 | 88.3 | 92.7 | 86.2 | 97.2 | 98.9 | 530.1 |
> | 20 | 66.8 | 88.3 | 92.7 | 86.2 | 97.2 | 98.9 | 530.1 |
> >>
> >Table: Image-text matching by re-ranking CLIP using different $\lambda$s on the Flickr30k dataset.
> | $\lambda$ | R@1(i2t) | R@5(i2t) | R@10(i2t) | R@1(t2i) | R@5(t2i) | R@10(t2i) | Total |
> | -- | :---: | :---: | :----: | :---: | :---: | :---: | :---: |
> | 0 | 65.4 | 87.2 | 91.7 | 85.4 | 97.1 | 98.7 | 525.5 |
> | 0.1 | 66.8 | 88.3 | 92.7 | 86.2 | 97.2 | 98.9 | 530.1|
> | 0.2 | 66.6 | 88.6 | 92.9 | 86.4 | 97.1 | 98.8 | 530.5|
> | 0.3 | 66.6 | 88.4 | 93.0 | 83.5 | 97.0 | 98.7 | 527.1|
>
>
> 2. What is the computation cost of the fine-tuned domain knowledge stage? F and M seem too large to make this step practical.
> > The computational cost is described as follows, which is not very high. On the Flickr30k and COCO datasets, the corresponding two models take about 3.5 hours and 12 hours for 25 training epochs on only one NVIDIA V100 GPU, respectively. Their batchsizes are both 128, which cost about 7.3GB GPU memory. The whole model has only one parameter matrix W, whose size is FxF, F=2048.
> >>
> >You mentioned M and N are the total numbers of words and regions in a minibatch of texts and images, respectively. As explained in page 4 lines 135-141, we also notice that original M and N could be large, so we use two effective strategies. 1) For the words, we only keep nouns, adjectives and verbs, and remove different tenses of verbs and the plural of nouns. So, a minibatch of texts have about 1024 words, i.e., M$\approx $1024. 2) For the images, we remove redundant regions belonging to same concepts in a similar way as non-maximum suppression. Thus, a minibatch of images totally have 2560 regions, i.e., N$\approx $2560. Therefore, directly training the model is feasible in practice.
>
> 3. Can the author provide the results which only use half the data of Visual Genome?
> > This is a good question. We have accordingly performed you mentioned experiment of dataset-based ablation study in Table 4 and Sections 4.6 (please see the table below for your convenience). The total number of annotated region-word pairs in Visual Genome is about 3.8M. We vary the number of used regions for computing a word-related prototype in {20, 50, 100, infinity}, and the corresponding total numbers of annotations are 0.56M, 1.39M, 2.78M, and 3.8M, which are 14.7%, 36.6%, 73.2%, and 100% of the whole annotations, respectively. From the table, we can find that even using only 14.7% annotations, the performance is still good.
> >>
> >Table: Image-text matching by ALBEF + MACK on the Flickr30k dataset. $f$ is the maximum number of used regions when computing a prototypical region presentation. $p$ is the percentage of how many word-region annotations of VG are used.
> | $f$  | $p$ | R@1(i2t) | R@5(i2t) | R@10(i2t) | R@1(t2i) | R@5(t2i) | R@10(t2i) | Total |
> | --| --| :--: | :---: | :---: | :---: | :---: | :---: | :---: |
> | 20   | 14.7% | 60.0 | 85.1 | 91.2 | 79.7 | 96.0 | 97.7 | 509.8 |
> | 50   | 36.6% | 60.2 | 85.3 | 91.1 | 79.9 | 96.0 | 97.7 | 510.3 |
> | 100  | 73.2% | 60.7 | 85.3 | 91.2 | 80.1 | 95.9 | 97.7 | 510.9 |
> | $\infty$| 100% | 61.8 | 85.8 | 91.4 | 80.1 | 96.4 | 97.7 | 513.3 |
>
> 4. If we want to study fine-grained structures within objects, the proposed method will fail. Also, the method will fail if no object associated with the word is detected.
> > Yes, our proposed knowledge-based method is still in its initial shape, which indeed has a lot of rooms for further improvement. Its effectiveness has been verified for the task of image-text matching, which focuses more on object/scene-level understanding rather than the mentioned structures insider objects. But we could generalize the knowledge-based idea from object-level (e.g., person, dog, cat) to part-level (e.g., head, hand, leg), which then might be useful for studying fine-grained structures within objects. In addition, the words in a text usually associate with salient objects in the images, which are easy to be detected in common cases. But for those images containing too many objects, it is possible that some word-related ones cannot be well detected.

---

> > ### Comment · Reviewer_NfVu · 2022-08-08
> > **Reviewer NfVu response**
> >
> > I appreciate the efforts made by the author and all the new results.Your rebuttal well address my concerns, and I raised my score from 4 to 6.

---

### Official Review · Reviewer_MQX3 · 2022-07-10

**Rating:** 5
**Confidence:** 4
**Soundness:** 2 fair
**Presentation:** 3 good
**Contribution:** 2 fair

**Summary:**

This paper proposes Multimodal Aligned Conceptual Knowledge (MACK) for image-text matching without requiring paired image-text for finetuning the pretrained model. In addition, the proposed approach can also be extended as a re-ranking method to improve the performance of current image-text matching approaches. It uses the word-region aligned datasets to get the pretrained general knowledge, and then finetune the model on unpaired image-text datasets to obtain the finetuned domain knowledge. During inference, image-text similarity scores are calculated by aggregating the word-region similarities. For the experiments, the pretrained general knowledge are obtained from Visual Genome dataset, and image-text matching experiments are conducted on COCO and Flickr30k datasets with both unpaired image-text setting and re-ranking setting.

**Questions:**

Please address my concerns in the weaknesses section.

**Limitations:**

The authors mentioned the limitations but do not address the potential negative societal impact of their work.

**Strengths And Weaknesses:**

Strengths:
1) This work proposes a new setup for image-text matching. It proposes to train or fine-tune image-text matching models without paired image text data.
2) It uses a pretrained detection model to provide the correspondences between visual and word features as domain knowledge.
3) Finetuning the pretrained domain knowledge on unpaired images and texts is effective for improving the image-text matching performance.
4) The proposed approach is not only validated on unpaired image-text matching settings, but also can be used as a re-ranking approach to improve the image-text matching performance of existing models such as CLIP [25] and ALBEF [16].

Weaknesses:
1) I am doubtful about whether the setup of unpaired image-text matching and the proposed solution is realistic or not. Firstly, although the training does not require ground-truth image-text correspondences, it requires the word-region alignment from the Visual Genome dataset to pretrained the detection model, and the ground-truth of VG dataset is more difficult and costly to annotate than image-text alignments. Secondly, the authors claim that they do not need paired image-text data for training, but the experiments are still conducted on COCO and Flickr30k image captioning datasets. Although the image-text correspondences are not used during training, there is an implicit prior that there exist one-to-one correspondences between images and text. It is unclear whether the proposed approach still works well on totally unpaired images and captions, e.g., Flickr images and COCO captions.
2) Why is the aggregation function in Eq.(4) of Section 3.3 defined as max-mean pooling? The ablation study experimented with mean-mean pooling, but what if it is max-max pooling or mean-max pooling?
3) The performance of unpaired image-text matching (Table 1) is quite low. And the authors do not provide the qualitative results on unpaired image-text matching. I am wondering if the retrieval results are reasonable with the unpaired setup.
4) No performance comparison is conducted. The authors could compare it to other unpaired matching or re-ranking approaches. If there are no previous approaches, at least it should be compared to some naive baselines, for example, training the unpaired image-text matching with cycle-consistency loss.
5) The cross-dataset re-ranking in Table 3 only provides marginal performance gain (less than 1%).
6) The re-ranking hyperparameters $k$ and $\lambda$ are not experimented in the ablation study.

---

> ### Author Response · Authors · 2022-08-02
> **Response to Reviewer MQX3**
>
> Thank you for spending so much time and effort on reviewing our paper, we will follow your valuable comments to improve our paper in the revision. Our responses to your concerns are as follows.
>
> 1.Whether the setup of unpaired image-text matching and the proposed solution is realistic or not. Firstly…the ground-truth of VG dataset is more difficult…Secondly…whether works well on totally unpaired images and captions, e.g., Flickr images and COCO captions.
> > For the first concern, it should be noted that using the word-region alignment is a commonly used setup in image-text matching. Since 2018 [13], almost all works [3,4,8,9,15,17,18,20,22,33,38] have followed this setup. Similar to them, our unpaired image-text matching is also based on the word-region alignment. But we try to avoid collecting and annotating large-scale image-text correspondences, which is under great demand in practical scenarios. For the second concern, we follow your suggestion to perform the training on the totally unpaired Flickr30k images and COCO captions, we find the results is slightly worse to previous results but still much better than pretrained knowledge. We will add these experiments and analysis in the revision.
> >>
> >Table: Unpaired image-text matching on the Flickr30k dataset.
> | Method | R@1(i2t) | R@5(i2t) | R@10(i2t) | R@1(t2i) | R@5(t2i) | R@10(t2i) | Total |
> | - | :---: | :---: | :--: | :--: | :--: | :--: | :--: |
> | Pretrained knowledge   | 10.8 | 26.6 | 35.8 | 4.1 | 10.8 | 18.0 | 106.2 |
> | Finetuned knowledge (Flickr30k image + Flickr30k captions) | 12.7 | 30.4 | 40.8 | 10.3 | 25.1 | 34.0 | 153.5 |
> | Finetuned knowledge (Flickr30k image + COCO captions) | 11.5 | 30.1 | 40.6 | 10.5 | 24.5 | 33.9 | 151.1 |
>
> 2. Why is the aggregation function defined as max-mean pooling? What if it is max-max pooling or mean-max pooling?
> > The motivation of using max-mean pooling is as follows. Given L words and N regions, we first compute their similarity matrix S (size: L x N). For each word, considering that the content of different regions could be noisy and redundant, we thus perform the max pooling on the column dimension to select a most related region. Then, to equally combine contributions of all the words to obtain the global image-text similarity, we perform the mean pooling on the row dimension. We compare all different kinds of pooling in the following table. We can see that the max-mean pooling performs the best.
> >>
> >Table: Unpaired image-text matching using different pooling operations on the Flickr30k dataset.
> | Method | R@1(i2t) | R@5(i2t) | R@10(i2t) | R@1(t2i) | R@5(t2i) | R@10(t2i) | Total |
> | --- | :---: | :---: | :---: | :---: | :---: | :---: | :---: |
> | mean-max pooling | 1.3 | 3.9 | 7.2 | 3.6 | 9.6 | 14.6 | 40.2 |
> | max-max pooling | 2.1 | 7.0 | 11.34 | 3.2 | 5.9 | 8.7 | 38.3 |
> | mean-mean pooling | 4.2 | 13.4 | 20.4 | 2.9 | 8.4 | 12.8 | 62.2 |
> | max-mean pooling | **10.8** | **26.6** | **35.8** | **4.1** | **10.8** | **18.0** | **106.2** |
>
> 3. I am wondering if the retrieval results are reasonable with the unpaired setup.
> > The major observation of image retrieval based on text query is that, for those incorrect images (with similar objects or scenes) in the retrieved top-10 results, most of which share at least one or two objects with the groundtruth matched images. We will give the corresponding results as supplementary materials in the revision.
>
> 4. It should be compared to some naive baselines.
> > We follow your suggestion to add the naive baseline by directly optimizing the cycle-consistency loss without knowledge in the following table. It seems that without using knowledge as cross-modal bridge or using image-text annotation, the model cannot well associate cross-modal features of regions and words.
> >>
> >Table: Unpaired image-text matching based on cycle-consistent losses on the Flickr30k dataset.
> | Method | R@1(i2t) | R@5(i2t) | R@10(i2t) | R@1(t2i) | R@5(t2i) | R@10(t2i) | Total |
> | - | :--: | :--: | :--: | :--: | :--: | :--: | :--: |
> | cycle-consistency loss (without knowledge) | 0.1 | 0.5 | 0.9 | 0.0 | 0.6 | 1.1 | 3.2 |
> | cycle-consistency loss (using knowledge) | **12.7** | **30.4** | **40.8** | **10.3** | **25.1** | **34.0** | **153.5** |
>
> 5. The cross-dataset re-ranking in Table 3 only provides marginal performance gain (less than 1%).
> > It should be noted that, for all the re-ranking results in Tables 2&3, we only use the pretrained general knowledge without the discriminative step of knowledge fine-tuning. Considering that the performance by re-ranked SOTA models is already high, it is promising to see that the simple pretrained general knowledge can achieve substantial improvements.
>
> 6. The re-ranking hyperparameters k and λ are not experimented in the ablation study.
> > Thank you for the reminder. You might find the detailed ablation study of two hyperparameters in the 1st question of Reviewer NfVu (he  raises the same concern). We will add these experiments in the revision.

---

### Official Review · Reviewer_Ypgg · 2022-07-11

**Rating:** 5
**Confidence:** 4
**Soundness:** 3 good
**Presentation:** 3 good
**Contribution:** 2 fair

**Summary:**

First of all, this paper pays attention to an important and challenging problem -- unpaired image/text retrieval and designs a framework that can be easily plugged into existing image-text matching models for re-ranking.

**Questions:**

see weeknesses.

**Ethics Review Area:**

["I don’t know"]

**Strengths And Weaknesses:**

S1: The proposed model is simple and can be easily plugged into existing image-text matching models to boost the performance.

S2: Extensive experiments are conducted and the results show that using the proposed MACK for re-ranking, the performance can be improved by around 1%.

W1: The proposed model actually employs human-annotated data to obtain prototypical region representations and prior knowledge and the dataset is composed of paired annotations, so "unpaired image-text matching" in the title is misleading.

W2: The pretrained general knowledge is highly related to the training data. In the paper, Visual Genome is used, but some other datasets can also be used, such as MSCOCO and ImageNet. I do not see any ablation studies on the pretrained general knowledge.

---

> ### Author Response · Authors · 2022-08-02
> **Response to Reviewer Ypgg**
>
> Thank you for spending so much time and effort on reviewing our paper, we will follow your valuable comments to improve our paper in the revision. Our responses to your concerns are as follows.
>
> 1. The proposed model actually employs human-annotated data to obtain prototypical region representations and prior knowledge and the dataset is composed of paired annotations, so "unpaired image-text matching" in the title is misleading.
> > We are sorry for the misleading. We would like to clarify your concern as follows. The “unpaired” mainly describes images and texts without pairwise annotations. While our used the annotations are from local image regions (e.g., objects) and words (e.g., nouns), which are usually used for the task of object detection, and also widely used by almost all image-text works since 2018 (the first one is [13]). Such a setup is also inspired by recent literates such as Unpaired Image Captioning via Scene Graph Alignments (ICCV2019), Jointly Pre-Training Transformers on Unpaired Images and Texts (arXiv2021), and Unpaired Vision-Language Pre-training via Cross-Modal CutMix (arXiv2022). We will refine our descriptions in the revision to avoid this issue.
>
> 2. The pretrained general knowledge is highly related to the training data. In the paper, Visual Genome is used, but some other datasets can also be used, such as MSCOCO and ImageNet. I do not see any ablation studies on the pretrained general knowledge.
> > Thanks for the comment. We actually have performed the dataset-based ablation study on the pretrained general knowledge in Table 4 and Section 4.6 (please see the table below for your convenience). In this experiment, we design four different datasets by varying the number of images linked to each word, and then use different numbers of images to compute the prototype representations to generate the knowledge. Our major observation is that: only using at most 20 images to compute a prototype representation for each word (now the total number of used annotations is only 14.7% of the whole annotations), the final results do not degenerate too much.
> >>
> > Table: Image-text matching by ALBEF + MACK on the Flickr30k dataset. $f$ is the maximum number of used regions when computing a prototypical region presentation. $p$ is the percentage of how many word-region annotations of VG are used.
> | $f$  | $p$ | R@1(i2t) | R@5(i2t) | R@10(i2t) | R@1(t2i) | R@5(t2i) | R@10(t2i) | Total |
> | -----| ----| :------: | :------: | :-------: | :------: | :------: | :-------: | :---: |
> | 20   | 14.7% | 60.0 | 85.1 | 91.2 | 79.7 | 96.0 | 97.7 | 509.8 |
> | 50   | 36.6% | 60.2 | 85.3 | 91.1 | 79.9 | 96.0 | 97.7 | 510.3 |
> | 100  | 73.2% | 60.7 | 85.3 | 91.2 | 80.1 | 95.9 | 97.7 | 510.9 |
> | $\infty$| 100% | 61.8 | 85.8 | 91.4 | 80.1 | 96.4 | 97.7 | 513.3 |
> >>
> > It is true that other datasets can also be used as alternatives. But you mentioned MSCOCO dataset might not be suitable since it has been used as one of our two benchmark datasets. While the ImageNet dataset has much smaller number of annotated objects than the selected Visual Genome dataset, which could largely lower the generalizability of the pretrained general knowledge. Compared with them, it is better to use a much smaller-scale dataset, which has a larger number of annotated objects but each one only has 20-30 annotated images. We will follow your comment to further improve our work.

---

### Official Review · Reviewer_iP3D · 2022-07-11

**Rating:** 7
**Confidence:** 5
**Soundness:** 4 excellent
**Presentation:** 4 excellent
**Contribution:** 4 excellent

**Summary:**

This paper studies the unpaired image and text retrieval task. Concretely, traditional image text retrieval model are learnt via paired and aligned image and text data. However, it is pretty common that the text is not following the same distribution as the text in the training set. Therefore, it is interesting and important to learn the model from unpaired image and text. This paper proposed an interesting model called MACK, which uses image patches to form prototypes for each word/phrase in the given text. Given a new image, it first detects all the objects in the image. Then it aligned the detected object with the prototypes. As the prototypes and the object representation live in the same space. Each prototype also links to one word. It can improve the image and text retrieval performance and conduct alignment across unpaired image and text.

**Questions:**

I like this idea a lot. Please address three questions mentioned above.

**Limitations:**

I don't see any negative societal impact.

**Strengths And Weaknesses:**

Pros:
I love this idea a lot! This is an elegant way to tackle the image and text retrieval task. The main challenging in this task is the semantic ambiguity in the text side. For example, all images of 'apple' with different appearance is aligned with the text 'apple'. How to align the text 'apple' with multiple 'apple' objects is a challenging task. This paper uses the prototypes to link the text and image worlds is very elegant.

Cons:
I have two high-level questions for the authors:
1. How to handle the out-of-vocabulary issue. For example, if the text has not been seen in the training set, how to handle this case? How to generate/obtain the prototypes for this OOV word/text?
2. How to handle the data imbalance problem? For example, some words might contain less images linked with it. This would lower the quality of the prototype representation. How to handle this case?
3. Correct me if I was wrong. To my understanding, the prototypes are the corresponding images representation of nouns. But how would the model distinguish between the following two cases: 'dog chases cat' vs. 'cat chases dog'. The prototypes for the 'dog' and 'cat' are all the same. How to use the prototypes to distinguish them?

Missing related work:
Zhang, B., Hu, H., Qiu, L., Shaw, P., & Sha, F. (2021). Visually Grounded Concept Composition. Findings of EMNLP. has a similar idea of the proposed approach.

---

> ### Author Response · Authors · 2022-08-02
> **Response to Reviewer iP3D**
>
> Thank you for spending so much time and effort on reviewing our paper, we will follow your valuable comments to improve our paper in the revision. Our responses to your concerns are as follows.
>
> 1. How to handle the out-of-vocabulary issue. For example, if the text has not been seen in the training set, how to handle this case? How to generate/obtain the prototypes for this OOV word/text?
> > This is a good question. To handle the OOV issue, we could extend the multimodal knowledge as an online version, so that it can adaptively enlarge its vocabulary by adding new knowledge related to the OOV words. In particular, to compute a visual prototype for an OOV word, we could leverage public resources (searching other datasets or the Internet) and obtain a certain number (computing a feasible prototype only needs about 20 images according to our experiments in Section 4.6) of related images. Then, we feed these images to the same image backbone network (i.e., faster-RCNN in our case) for representation extraction, and average their representations as the desired prototype representation. At last, the paired prototype-OOV word are regarded as new knowledge, which will be added to the multimodal knowledge and thus can handle the out-of-vocabulary words.
>
> 2. How to handle the data imbalance problem? For example, some words might contain less images linked with it. This would lower the quality of the prototype representation. How to handle this case?
> > We also noticed this problem and accordingly performed the experiment of dataset-based ablation study in Table 4 and Sections 4.6 (please see the table below for your convenience). We find that when the maximum number of used images for computing a prototype representation is set as 20 (i.e., each word only links to at most 20 related images), we can still obtain satisfying results. And, allowing some words using more than 20 images (i.e., 50, 100, infinity) does not necessarily lead to significant better performance. This indicates that the prototype representations might be not very sensitive to the data imbalance problem. Although there is still a number of words having only a few linked images (i.e., < 5), we might supplement them by regarding them as query on the Internet to find more linked images. We will provide more discussions in the revision.
> >>
> >Table: Image-text matching by ALBEF + MACK on the Flickr30k dataset. $f$ is the maximum number of used regions when computing a prototypical region presentation. $p$ is the percentage of how many word-region annotations of VG are used.
> >| $f$  | $p$ | R@1(i2t) | R@5(i2t) | R@10(i2t) | R@1(t2i) | R@5(t2i) | R@10(t2i) | Total |
> | -----| ----| :------: | :------: | :-------: | :------: | :------: | :-------: | :---: |
> | 20   | 14.7% | 60.0 | 85.1 | 91.2 | 79.7 | 96.0 | 97.7 | 509.8 |
> | 50   | 36.6% | 60.2 | 85.3 | 91.1 | 79.9 | 96.0 | 97.7 | 510.3 |
> | 100  | 73.2% | 60.7 | 85.3 | 91.2 | 80.1 | 95.9 | 97.7 | 510.9 |
> | $\infty$| 100% | 61.8 | 85.8 | 91.4 | 80.1 | 96.4 | 97.7 | 513.3 |
>
> 3. To my understanding, the prototypes are the corresponding images representation of nouns. But how would the model distinguish between the following two cases: 'dog chases cat' vs. 'cat chases dog'. The prototypes for the 'dog' and 'cat' are all the same. How to use the prototypes to distinguish them?
> > This is a very good point. The mentioned case is also studied in a previous paper (i.e., learning semantic concepts and order for image and sentence matching, CVPR2018) as the problem of semantic order learning. To be honest, our current method cannot well distinguish different semantic orders, since it mainly focuses on cross-modal matching in the level of local region-word. But a straightforward solution might be computing the prototypes not only for the nouns and adjectives, but also for the verbs (e.g., chases in your mentioned case), and then aggregate all the prototypes in a sequential order to get the whole image presentation for global image-text matching. Thus, the different scene-level semantics of 'dog chases cat' or 'cat chases dog' can be better described and distinguished.
>
> 4. Missing related work: Zhang, B., Hu, H., Qiu, L., Shaw, P., & Sha, F. (2021). Visually Grounded Concept Composition. Findings of EMNLP. has a similar idea of the proposed approach.
> > Thanks for the reminder. We will check and add more related works.

---

> > ### Comment · Reviewer_iP3D · 2022-08-10
> > **Post rebuttal comments**
> >
> > Thanks authors for the detailed comments. My questions have been resolved. As I said, I like this paper a lot. I would maintain my ratings.

---

### Meta-Review · Area_Chair_ys9a · 2022-08-20

**Recommendation:** Accept
**Confidence:** Certain

**Metareview:**

This paper proposes MACK for unpaired image-text matching, which can also be extended as a re-ranking method to improve the performance of current image-text matching approaches. It uses the word-region aligned datasets to get the pretrained general knowledge, and then finetune the model on unpaired image-text datasets to obtain the finetuned domain knowledge. During inference, image-text similarity scores are calculated by aggregating the word-region similarities.

The authors have done a good job during rebuttal. After rebuttal, the paper received scores of 5567, which is in general positive. This work proposes a new setup for image-text matching, and the studied direction seems novel. The proposed approach is not only validated on unpaired image-text matching settings, but also as a re-ranking approach to improve the performance of existing models such as CLIP and ALBEF.

On the other hand, clear limitations also exist, such as (1) the current method is limited to the visual properties annotated in Visual Genome and the ability of the pre-trained Faster-RCNN; (2) whether the setup of unpaired image-text matching and the proposed solution is realistic or not; and (3) the performance of unpaired image-text matching itself (Table 1) is quite low.

On balance, the merits slightly outweigh the flaws, and the AC would like to recommend acceptance of the paper.

**Award:**

No

---

### Decision · Program_Chairs · 2022-09-14

Accept